# Tumour- and Non-Tumour-Associated Factors That Modulate Response to PD-1/PD-L1 Inhibitors in Non-Small Cell Lung Cancer

**DOI:** 10.3390/cancers17132199

**Published:** 2025-06-30

**Authors:** Maryam Khalil, Ming-Sound Tsao

**Affiliations:** 1Princess Margaret Cancer Centre, University Health Network, Toronto, ON M5G 2C4, Canada; maryam.khalil@uhn.ca; 2Department of Laboratory Medicine and Pathobiology, University of Toronto, Toronto, ON M5S 3K3, Canada; 3Department of Medical Biophysics, University of Toronto, Toronto, ON M5S 2C4, Canada

**Keywords:** PD-1, PD-L1, immune checkpoint inhibitors, NSCLC, patient response

## Abstract

The immune system plays a key role in regulating the growth of tumour cells. Mobilizing the immune system with PD-1/PD-L1 inhibitors has shown clinical benefits in subsets of patients with lung cancer. Currently, PD-L1 tumour expression is the only validated predictive biomarker for PD-1/PD-L1 blockade therapy in these patients. Advancing predictive and prognostic biomarkers will enable personalized immunotherapy approaches and maximize patient benefit. This review aims to summarize the tumour- and non-tumour associated factors that affect the response to immune checkpoint inhibitors.

## 1. Introduction

Immunotherapy targeting immune checkpoint proteins, especially PD-1/PD-L1, has become an increasingly important treatment strategy for various malignancies; it has especially changed the treatment paradigm for patients with non-small cell lung cancer (NSCLC). In subsets of patients, PD-1/PD-L1 inhibitors have shown substantial clinical efficacy, leading to durable or complete responses.

PD-1 (CD279), a co-inhibitory receptor expressed on the surface of immune cells, belongs to the immunoglobulin superfamily that was first identified and characterized in 1992 in mice [1]. The PD-1 receptor binds to two ligands, programmed death ligand 1 (PD-1/B7-H1/CD274) and programmed death ligand 2 (PD-L2/B7-DC/CD273). PD-1 signalling can inhibit T cell functions by promoting T cell exhaustion and suppression [2,3]. Blockade of either PD-1 or its ligands has shown consistent immune-potentiating effects in preclinical models. Antibodies against PD-1 or PD-L1 can enhance or restore T cell effector function, including cytolytic activity against tumour cells [4,5,6].

Immune checkpoint inhibitors (ICIs) have demonstrated benefit in a broad population of patients with NSCLC. Current approved ICIs targeting PD-1/PD-L1 in NSCLC are highlighted in Table 1. These ICIs have been approved as a standard of care for patients with advanced NSCLC whose tumours progress on first-line cytotoxic systemic therapy. ICIs, including nivolumab [7,8], pembrolizumab [9], and atezolizumab [10], have shown improvement in overall survival (OS) compared to treatment with docetaxel. Nivolumab was the first PD-1 inhibitor tested in phase I trials [11,12] for clinical activity and safety evaluation. Checkmate-003 evaluated nivolumab in 296 patients with advanced solid tumours, including NSCLC, and showed an 18% response rate in NSCLC patients [11]. However, not all patients respond to the treatment, and the mechanism behind the resistance to these antibodies is poorly understood.

The expression of PD-L1 detected by immunohistochemistry (IHC) is currently the most widely accepted predictive biomarker of the response to PD-1/PD-L1 inhibitors [13,14,15]. In advanced-stage NSCLC patients, the response rates to these ICIs partly correlate with the level of PD-L1 expression, and patients with a high PD-L1 tumour proportion score (TPS) have shown greater response to ICIs versus patients who have a low TPS. The superior clinical efficacy of PD-1 inhibitor pembrolizumab vs. chemotherapy is seen mainly in patients with high PD-L1-expressing tumours (TPS ≥ 50%) [16,17]. These results set a new standard of care as a first-line treatment for patients with advanced NSCLC. Multiple reviews [18,19,20,21] have previously delved into the impact of PD-L1 expression as a predictive biomarker for immunotherapy response, and, therefore, it will not be a focus of this review. However, it is generally recognized that PD-L1 expression is an imperfect predictor of the response to ICI due to tumour heterogeneity and the inducible nature of PD-L1.

This review summarizes the current understanding of tumour-associated and non-tumour-associated factors, in addition to PD-L1 expression, that may influence response to PD-1/PD-L1 inhibitors. Here, we focus on the factors that affect the activation and suppression of the immune response and determine how these factors predict response to immunotherapy.

## 2. Tumour-Associated Factors

### 2.1. STK11/LKB1 Mutations

The Serine Threonine Kinase 11 (*STK11*) gene is located in the telomeric region of the short arm of chromosome 19. It is a commonly mutated gene in human lung cancers, with at least 10–35% of NSCLC cases harbouring this mutation [22,23]. Additionally, 18% of lung adenocarcinoma (LUAD) [24] and 5% of lung squamous carcinoma (LUSC) [25] patients have mutations in *STK11*. The *STK11* gene encodes for liver kinase B1 (LKB1), which is a tumour suppressor that activates AMP-activated protein kinase (AMPK). Along with LKB1, AMPK is a central metabolic regulator that controls glucose and lipid metabolism in response to nutrient changes and energy levels. AMPK is a heterotrimer that is activated when the intracellular levels of ATP decrease, and AMP increases in conditions such as hypoxia or nutrient deprivation [26,27].

One of the major pathways controlled by LKB1-AMPK is mTOR, which controls cell growth in eukaryotes and is poorly regulated in most human cancers. mTOR Complex 1 (mTORC1) recruits various downstream substrates, such as MYC, Hypoxia-Inducible Factor 1 (HIF1), and cyclin D, which promote processes such as angiogenesis, cell growth, and cell cycle progression, all of which play a role in tumour progression [28]. Additional upstream components of mTORC1 include tuberous sclerosis complexes 1 and 2 (TSC1/2). TSC2 is a tumour suppressor that forms a heterodimer complex with TSC1 and indirectly inhibits mTORC1. Thus, the loss of TSC1/TSC2 leads to the hyperactivation of the mTORC1 complex [29]. Low levels of energy/ATP within the cell result in LKB1-dependent activation of AMPK, which promotes the production of rapidly available energy with ATP and slows down cell growth via the phosphorylation of TSC2 and the inhibition of mTORC1 activity [30,31]. Hence, LKB1 negatively regulates mTORC1, and LKB1 inactivation can lead to mTORC1 hyperactivation (Figure 1).

Several cancer types have shown the loss of LKB1 function through somatic alterations in the *STK11* gene. A study analyzing tissue from 4446 cancer patients found that 1.35% had *STK11* alterations either in the tissue, ctDNA, or both. In the *STK11*-mutated cohort, 45% were patients with NSCLC, the largest subgroup in the cohort [32]. In localized, resected NSCLC, *STK11* mutational status was evaluated in 352 patient samples. The study found that patients harbouring mutations in *P53*, *STK11*, and *SMARCA4* had the worst overall survival (OS) outcome, with a hazards ratio of 1.66 (95% CI 1.05–2.61) for *STK11* mutations [33].

In recent studies, the loss of LKB1 has been shown to affect the tumour immune microenvironment. To show the impact of LKB1 mutations on immune modulation, an LKB1-deficient murine LKR13 cell line was generated using CRISPR/Cas9; these cells were implanted in syngeneic mouse models, which showed lower numbers of CD3^+^CD8^+^ T lymphocytes compared to the wild type [34]. In addition to affecting tumour infiltration, LKB1 is crucial for metabolic pathways in T cell progenitors, and its loss has been shown to inhibit thymocyte differentiation and, in turn, the production of T lymphocytes [35,36]. An evaluation of T cell function in LKB1-deficient tumours revealed significantly less IFNg and Ki-67 expression in CD4^+^ and CD8^+^ tumour-infiltrating T cells [37]. Furthermore, the loss of LKB1 expression due to a single allele mutation in STK11 in T cells leads to the formation of gastrointestinal polyps in mice due to the inflammatory cytokines from the mutated T cells [38]. The inactivation of LKB1 also promotes the production of pro-inflammatory cytokines, including IL-6 and CXCL7, which leads to the accumulation of T cell suppressor neutrophils with low infiltration of CD4^+^/CD8^+^ T lymphocytes and lower PD-L1 expression [37]. The loss of LKB1 has also been shown to result in the inactivation of stimulator of interferon genes (STING). STING mediate the expression of type I interferon and various chemokines. One study found that STING inactivation resulted in the loss of chemokines, including CCL5 and CXCL10, which play a role in the recruitment of T cells [39].

The impact of LKB1 mutations on the immune microenvironment has also been studied in NSCLC patients by in-depth immune profiling of 221 untreated resected tumours of patients with lung adenocarcinoma. Tumours with LKB1 mutations were characterized by lower PD-L1 expression, and reduced CD8^+^ T cell and dendritic cell density. In contrast, tumours with *TP53* mutations had higher levels of CD8^+^ T cells, indicating an inflamed tumour microenvironment. Additionally, the *TP53/STK11* co-mutation cohort was significantly associated with lower CD8^+^ T cell density and reduced PD-L1 expression [39], with different mutational subsets exhibiting different patterns in the tumour microenvironment. Tumours with *KRAS/LKB1* co-mutations expressed lower levels of immune markers, including PD-L1. Conversely, tumours with *KRAS/TP53* mutations had greater T cell infiltration and higher expression of PD-L1, indicating a strong adaptive immune response [40]. *STK11* mutations were also associated with a lack of PD-L1 expression [34,37] and were enriched for negative PD-L1 staining [41,42], whereas tumours with high PD-L1 (>50%) were less frequently mutated for *STK11* [43]. Thus, tumours harbouring *STK11* mutations can be classified as immunologically “cold” and predictive of a lack of clinical benefit from immune checkpoint inhibitors.

The impacts of *LKB1* mutations have been evaluated in several retrospective studies. Skoulidis et al. [34] assessed an MDACC cohort with 66 non-squamous NSCLC patients treated with PD-1/PD-L1 inhibitors with available genomic profiling and PD-L1 expression. *LKB1* mutations were associated with shorter OS (HR 14.3; 95% CI, 3.4–50.0, *p* < 0.0001, log rank test) and PFS (HR 4.76; 95% CI, 2.0–11.1, *p* < 0.00012, log rank test) in patients treated with anti-PD-1/PD-L1, irrespective of PD-L1 expression levels. A pooled analysis from the OAK and POPLAR trials, as well as a study by Rizvi et al., showed similar results [42,44]. Overall, alterations in LKB1 were shown to be associated with a lack of benefit from immune checkpoint inhibitors. However, it is important to note that POPLAR is an exploratory retrospective phase 2 study and needs additional validation.

Preclinical and early clinical studies identify several combination approaches that can overcome LKB1-driven ICI resistance. For instance, dual CTLA-4 and PD-(L)1 blockade restores T cell responses in *STK11*-mutant tumours by reprogramming the myeloid compartment toward anti-tumour phenotypes and restoring anti-tumour efficacy [45]. Furthermore, the depletion or functional blockade of granulocytic MDSCs in LKB1-deficient tumours reverses immune suppression and synergizes with PD-1 blockade to enhance T cell response and inhibit tumour growth [46]. Strategies to reactivate deficient STING/type I interferon pathways also show promise since the loss of tumour STING signalling can impair T lymphocyte infiltration. STING activation has been shown to restore tumour immunogenicity and T cell priming in *STK11*-mutant tumours [47]. Together, these multi-faceted strategies, guided by biomarker-driven patient selection for *STK11* status, present viable paths to overcome the loss of immunogenicity caused by LKB1 loss.

Currently, there are no guidelines that recommend the systematic evaluation of *LKB1* mutational status in NSCLC, and current practice supports the use of ICI in first-line settings as well as in all advanced NSCLC patients without oncogenic driver mutations, regardless of *LKB1* status.

### 2.2. KEAP1 Mutations

Kelch-like ECH-associated protein 1 (KEAP1) is a metalloprotein that contains 27 cysteine residues, but only the conserved Cys23, Cys151, Cys273, and Cys278 residues have been shown to play a functional role in KEAP1 activity. The expression of recombinant KEAP1 in cell lines has shown that mutants missing Cys23, Cys273, and Cys278 cannot regulate NRF2 activity [48]. KEAP1 is a component of the cullin 3 (CUL3)-based E3 ubiquitin ligase complex, which controls the stability of NF-E2-related factor 2 (NRF2). NRF2 is a transcription factor that regulates the antioxidant response in cells and is sensitive to electrophilic and oxidative stresses, such as reactive nitrogen species (RNS) and reactive oxygen species (ROS). In the nucleus of cells, NRF2 transcriptionally activates antioxidant genes by binding to antioxidant response elements (AREs) or electrophile response elements (EpREs). NRF2 is repressed directly by KEAP1, making the KEAP1/NRF2 pathway crucial in regulating oxidative stress and damage. Under normal cellular conditions, NRF2 levels are very low, but exposure to various stressful stimuli, such as RNS or ROS, can lead to dramatic changes in the levels of this protein [49]. Recent studies have also shown that the KEAP1/NRF2 pathway can play a role in cancer progression, metastasis, and resistance to therapy.

In addition to redox balance, Nrf2 activates genes involved in metabolism, xenobiotic response, and cell survival. Oxidative stress caused by ROS-mediated cell damage is detrimental to cell survival and can contribute to tumourgenesis [50]. The activation of the Nrf2 (Figure 2) pathway leads to not only the scavenging of ROS but also the upregulation of multiple drug-metabolizing enzymes, such as GCL, AKR, MRPs, and UGT. The downstream targets of Nrf2 are categorized into three classes: phase I and II drug metabolizing enzymes and phase III drug transporters. Phase I and phase II enzymes are responsible for oxidizing, reducing, hydrolyzing, and metabolizing xenobiotics, carcinogens, and drugs, whereas phase III drug transporters remove drugs, xenobiotics, and metabolites from cells [51]. Nrf2, therefore, is a key player that functions not only to prevent tumourgensis in normal cells by redox balance, but it can also induce resistance to drugs by enhancing drug metabolism and extrusion from cells. This dual role of Nrf2 is key to understanding how mutations in KEAP1 can promote resistance to ICI.

Several factors lead to the aberrant activation of NRF2 in cancerous cells. Somatic mutations in the *KEAP1*, *NRF2*, or *Cul3* genes [53,54]; the epigenetic silencing of the *KEAP1* gene [55]; KEAP1 cysteine modifications [56]; and the accumulation of KEAP1-interacting proteins, such as p62 [57] and p21 [58], may positively regulate NRF2 expression by affecting KEAP1 activity. Mutations involving the *KEAP1/Nrf2/Cul3* genes have been reported in 23% of lung adenocarcinomas (LUADs) [59] and 34% of lung squamous carcinomas (LUSCs) [60]. These mutations have also been found in the histologically normal airway epithelium in early-stage NSCLC patients, highlighting the potential role of KEAP1 as a driver [61]. The overactivation of Nrf2 caused by *KEAP1* mutations has been associated with radiation and chemotherapy resistance, largely due to the drug-metabolizing and transporting functions of Nrf2 [62]. NRF2-driven metabolic reprogramming supports anabolic processes and survival under nutrient limitation, and NRF2 addiction in tumours may promote tumour cell proliferation, metastasis, and poor prognosis. In NSCLC, *KEAP1* mutations have been correlated with aggressive disease and therapy resistance [63,64]. Similarly, in head and neck squamous cell carcinoma, KEAP1 loss leads to the nuclear accumulation of NRF2 and enhances xenobiotic metabolism enzymes, causing resistance to chemotherapy. Controlling NRF2 activity by inducing KEAP1 expression or silencing NRF2 may increase sensitivity to chemotherapeutic agents in *KEAP1*-mutated cells [65].

Since NRF2 influences both innate and adaptive immunity, *KEAP1* mutations may also play a significant role in modulating the tumour microenvironment. Under normal conditions, NRF2 limits excessive inflammation by suppressing pro-inflammatory cytokine production in macrophages and other immune cells [66]. However, in tumours, NRF2 activation may reprogram tumour-associated macrophages (TAMs) to be pro-tumourigenic [67] and inhibit the anti-tumour response of CD8^+^ T cells [68]. Additionally, Zavitsanou et al. generated an antigenic lung cancer mouse model and showed that tumours harbouring *KEAP1* mutations accelerated tumour growth and decreased dendritic cell and T cell responses, which promoted resistance to immunotherapy. This was validated in a cohort of 19 LUAD patients who showed diminished dendritic cell and T cell infiltration [69]. Additionally, in a cohort of 543 LUAD patients, with 96 (17.6%) patients harbouring *KEAP1* mutations, Cheng et al. [70] showed that six immune cell types, including CD8^+^ T cells, CD4^+^ T cells, B cells, macrophages, dendritic cells, and neutrophils, showed markedly lower infiltration levels in *KEAP1*-mutant tumours as compared to *KEAP1* wild types [70]. The characterization of the tumour immune microenvironment of *KEAP1*-mutant tumours in a K1P mouse model showed impaired expansion of CD11c^+^ immune cells, further emphasizing the immunosuppressive effects of *KEAP1* mutations [71]. Surprisingly, in vivo studies show higher expression of PD-L1 in CD8^+^T cells in *KEAP1^null^/KRAS^G12D^*-mutant tumours as compared to *KEAP1^WT^/KRAS^G12C^* tumours [72], suggesting the role of *KEAP1* in regulating the immune response and immunosuppression.

Despite the preclinical evidence, the clinical impact of *KEAP1/NRF2* mutations on the efficacy of immunotherapy in NSCLC remains controversial. Studies have shown associations between *KEAP1* mutations and poor responses to immune checkpoint blockades (Table 2). The association between *KEAP1* mutation and PD-L1 upregulation observed in preclinical studies has not been observed in clinical trial samples, suggesting an alternate mechanism of immune evasion. Patients with *KEAP1/NRF2* mutations have been found to have an inferior mOS when compared to control groups after immunotherapy treatment. The response in patients with *KEAP1*-mutated tumours has also been studied based on clonality. In two independent cohorts of patients (n = 237 and n = 461) treated with anti-PD-1/PD-L1, *KEAP1* clonal inactivation and the loss of heterozygosity (KEAP1 C-LOH) showed shorter PFS and OS compared to wild-type patients, whereas no significance was observed between patients with *KEAP1* subclonal mutations or partial inactivation and those with wild-type tumours [73]. These findings indicate the possibility of different KEAP1 mutational classes with distinct clinal outcomes and immunophenotypic features.

*KEAP1* co-mutations with either *STK11* or *KRAS* identify NSCLC patients with particularly poor prognosis and limited response to treatment. In a multicenter cohort of advanced lung adenocarcinoma patients treated with PD-(L)1 blockade, those with concurrent *STK11* and KEAP1 mutations had lower objective response rates and shorter PFS and overall survival compared to patients with wild-type or single-mutant tumours [74]. Gene expression analyses of the dual-mutant tumours revealed an immune “cold” phenotype characterized by reduced T cell infiltration and the downregulation of immune-related pathways, providing a clinical correlation for the observed resistance to immunotherapy [74]. In *KRAS*-mutant NSCLC, the co-mutation of *KEAP1* confers independent adverse prognosis and diminished benefits from both chemotherapy and ICIs. In a cohort of 330 patients with advanced *KRAS*-mutant NSCLC, KEAP1/NFE2L2 alterations were associated with shorter overall survival (HR 1.96; 95% CI, 1.33–2.92; *p* ≤ 0.001), reduced duration of response to platinum-based chemotherapy (HR 1.64; 95% CI, 1.04–2.59; *p* = 0.03), and decreased survival after initiation of PD-(L)1 inhibition (HR 3.54; 95% CI, 1.55–8.11; *p* = 0.003 [75].

**Table 2 cancers-17-02199-t002:** Clinical trials reporting efficacy of PD-1/PD-L1 inhibitors in NSCLC patients with KEAP1 mutations and the effect of KEAP1 mutations on PD-L1 expression and TMB status.

Clinical Trial	Patient Number	Trial Design	Results	Reference
**Phase 3 Studies**				
KEYNOTE-042 (NCT02220894)	KEAP1^mut^: 64 KEAP1^WT^: 365	Pembrolizumab vs. platinum-based chemotherapy	No significant difference observed in KEAP1-mutated vs. WT NSCLC pateints. Pembrolizumab: 17 vs. 16.9 months; platinum-based chemotherapy: 8.9 vs. 12.2 months	[76]
MYSTIC (NCT03873246)	KEAP1^mut^: 169 KEAP1^WT^: 943	Durvalumab vs. Durvalumab plus Tremelilumab vs. chemotherapy	Poor OS was observed in KEAP1-mutated vs. WT NSCLC pateints. Durvalumab alone: 7.6 vs. 14.6 months; Durvalumab + Tremelilumab: 9.2 vs. 11.3 months; Chemotherapy: 6.3 vs. 13.3 months.	[77]
OAK (NCT02008227)	KEAP1^mut^: 90 KEAP1^WT^: 508	Atezolizumab vs. Docetaxel	In high-PD-L1 NSCLC patients, OS in the KEAP1^mut^ group was 6.24 months vs. 22.47 months in the KEAP1^WT^ group (*p* = 0.003).	[78]
**Pooled Studies/Analysis**				
MSKCC—Database Analysis	KEAP1^mut^: 69 KEAP1^WT^: 202	Immune checkpoint inhibitors	Patients with KEAP1 mutations had shorter OS than WT pateints (*p* = 0.040)	[79]
OAK and POPLAR (NCT01903993)	KEAP1^mut^: 90 KEAP1^WT^: 508	Atezolizumab vs. Docetaxel	In both the atezolizumab and docetaxel groups, KEAP1-mutated patients had worse survival than WT patients, at 7.06 vs. 16 months (*p* < 0.001) and 6.14 vs. 10.81 months (*p* < 0.001), respectively	[42]
IRE	KEAP1^mut^: 16 KEAP1^WT^: 72	Chemotherapy	KEAP1-mutated pateints showed worse OS (HR = 1.8 95% CI:1.03–3.13, *p* = 0.037) and PFS (HR = 2.09 95% CI:1.20–3.65, *p* = 0.009) as compared to WT patients.	[80]
MSKCC	KEAP1/Nrf2^mut^: 365 KEAP1/Nrf2^WT^: 4272	Analysis of the relationship between KEAP1/NRF2 mutations, TMB, and OS	The KEAP1-mutated cohort had higher PD-L1 expression and TMB; the median OS in the KEAP1-mutated cohort was worse (11.5 vs. 22.3 months, *p* < 0.01)	[81]

Overall, the current results (Table 2) suggest that the KEAP1/NRF2 mutation is a poor prognosis. Depending on the context, the Nrf2-KEAP1 pathway can harbour tumour-suppressor or oncogenic properties. However, the mechanisms of resistance to anti-PD-1/PD-L1 remain unclear, and more studies are needed to characterize the immune microenvironment of KEAP1-mutant tumours.

### 2.3. EGFR Mutations

Epidermal growth factor receptor (*EGFR*), also known as ErbB1/HER1, is a member of the *EGFR* family that also includes ErbB2/HER2, ErbB3/HER3, and ErbB4/HER4. *EGFR* is a receptor tyrosine kinase (RTK), commonly upregulated in NSCLC [82]. The majority of mutations/truncations in EGFR stabilize ligand-independent dimerization with ERBB family members to promote constitutive activation of the receptor [83]. Additionally, T790M mutations and insertion mutations in exon 20 of the kinase domain also lead to an increase in *EGFR* phosphorylation levels and confer resistance to tyrosine kinase inhibitors [84].

NSCLC patients with *EGFR* mutations respond well to tyrosine kinase inhibitors (TKIs) with minimal adverse effects; however, the majority will develop resistance, emphasizing the need for additional treatment strategies [85]. The utility of anti-PD-1/PD-L1 immunotherapy in *EGFR*-mutated tumours has been investigated in several trials (Table 3), which consistently showed that NSCLC tumours with *EGFR* mutations have a poor response to these therapies. The mechanisms mediating this poor response remain unclear.

Multiple studies have reported that *EGFR*-mutant lung tumour tissues have a higher expression of PD-L1 more frequently than *EGFR* WT tumours [86,87,88,89]. In contrast, other studies have shown either lower expression or no association of PD-L1 in *EGFR*-mutant NSCLC [90,91,92]. A pooled analysis of 15 clinical studies, with an integrated analysis of the mutation profile and PD-L1 expression, revealed that patients with *EGFR* mutations had lower PD-L1 expression [93]. These findings were also confirmed in the Guangdong Lung Cancer Institute (GCLI) and The Cancer Genome Atlas (TCGA) validation cohorts in the same study [93]. The inconsistency suggests the PD-L1 expression may not be the primary factor for the lack of response of patients with EGFR-mutant NSCLC to PD-1/PD-L1 inhibitor therapies.

EGFR pathway activation is also associated with tumour-promoting inflammatory cytokines and increased markers of T cell exhaustion in EGFR-driven mouse models of lung cancer [87]. *EGFR*-mutant tumours also secrete elevated IL-6, which impairs T and NK cell activity, further skewing the microenvironment toward immunosuppression [94]. These changes correlate with reduced CD8^+^ TIL density and dysfunctional effector phenotypes in EGFR-driven cancers [95]. Activating *EGFR* mutations may also upregulate TGF-β via EGFR–ERK1/2–p90RSK signalling, which could reduce CD8^+^ T cell infiltration and effector function in NSCLC, fostering a cold TME and resistance to PD-1 blockade [96]. Notably, PD-1 inhibitors showed an improved survival rate in mouse models with *EGFR* mutations [87], in contrast to results from clinical trials [8,9,10,97]. The disparity indicates a complex relationship between TME, *EGFR* mutations, and response to ICIs.

In major NSCLC clinical trials investigating PD-1/PD-L1 inhibitors, only 5–14% of the total recruited patients had *EGFR* mutations. Since these trials were not designed to study the impact of ICIs in *EGFR*-mutant tumours, treatment efficacy in the *EGFR*-mutated group was studied by patient subgroup analysis (Table 3). No clinical benefit was observed in patients with *EGFR*-mutant NSCLC with anti-PD-1/PD-L1 inhibitors.

The efficacy of pembrolizumab in TKI-naïve advanced NSCLC patients with *EGFR* mutations and high PD-L1 tumour expression was evaluated in a phase II clinical trial. The planned accrual of the trial was 25 patients; however, after 11 patients who received pembrolizumab did not show a favourable response, the trial enrollment was halted. None of the 11 patients with *EGFR* mutations responded [98].

Increased T cell infiltration and an immunogenic tumour immune microenvironment have been shown to correlate with a favourable response to immunotherapy. *EGFR* TKIs lead to immunogenic apoptosis of cancer cells and cause the release of neoantigens, which can recruit T cell antigen presentation. However, the combination of anti-PD-1/PD-L1 with *EGFR* TKIs did not show a greater response in NSCLC patients with *EGFR* mutations. The combination of gefitinib (n = 7) and erlotinib (n = 12) with pembrolizumab was evaluated in a phase I/II KEYNOTE-021 trial in advanced NSCLC patients with *EGFR* mutations. The pembrolizumab plus gefitinib combination resulted in grade 3/4 liver toxicity in five out of seven patients, leading to treatment discontinuation. While the adverse effects from pembrolizumab plus erlotinib were similar to erlotinib monotherapy, pembrolizumab plus erlotinib failed to improve ORR compared to previous monotherapy studies [99]. Another phase I open-label clinical trial conducted in 56 *EGFR* TKI-naïve NSCLC patients observed higher toxicity profiles in patients receiving the combination treatment than in those receiving either drug alone. Furthermore, no significant improvement was observed in ORR or PFS compared to the previously reported gefitinib monotherapy in a similar patient population [100]. Additionally, preclinical studies have shown that Osimertinib can enhance the anti-tumour effects of anti-PD-1/PD-L1 therapy by increasing CD8^+^ T cell infiltration in the tumour [101]. However, the phase Ib TATTON trial, which evaluated the safety and tolerability of combining durvalumab and osimertinib in *EGFR*-mutant NSCLC patients who progressed after previous EGFR-TKI, demonstrated that 38% of the patients developed serious interstitial lung disease, and the combination was deemed unfeasible [102].

However, the response to ICI treatment varies depending on the type of *EGFR* mutation in NSCLC patients. A retrospective study analyzed clinical data from 171 NSCLC patients with *EGFR* mutations that were treated with anti-PD-1/PD-L1 alone or in combination with anti-CTLA-4. The presence or absence of T790M *EGFR* mutations did not show benefits in response to immune checkpoint inhibitors [103]. Patients with *EGFR* L858R mutations and exon 19 deletions exhibited reduced benefit from ICIs as compared to patients with WT *EGFR*; however, patients with L858R mutations showed a better response than those with exon 19 deletions (ORR, 22% in the *EGFR* WT group, 16% in the L858R group, and 7% in the exon 19 deletion group). Interestingly, tumours with exon 19 *EGFR* deletions exhibited lower TMB than the L858R subgroup [103]. Moreover, uncommon *EGFR* mutations, including exon 20 insertions, G719X, S768I, and L861Q, were associated with a favourable response in Chinese NSCLC patients to anti-PD-1/PD-L1. This response was associated with high PD-L1 expression and CD8^+^ TILs in the tumour microenvironment [104]. A Japanese retrospective study (n = 27) of PD-1 inhibitors concluded that NSCLC patients with uncommon *EGFR* mutations and without T790M mutations exhibited significantly longer PFS than patients with common mutations or T790M mutations in *EGFR* [105]. These results indicate the possibility of the personalized use of ICIs by screening for *EGFR* mutations in NSCLC patients; however, further studies may be warranted.

**Table 3 cancers-17-02199-t003:** Clinical trials reporting efficacy of PD-1/PD-L1 inhibitors in NSCLC patients with EGFR mutations.

Clinica Trial	Number of Participants	Trial Design	Results	Reference
KEYNOTE-010 (Phase III—NCT01905657)	EGFR mutant = 86 EGFR WT = 875	Pembrolizumab vs. Docetaxel	No OS benefit was shown with pembrolizumab over docetaxel in NSCLC patients with EGFR mutations (HR = 0.88 95% CI:0.45–1.70) vs. WT (HR = 0.66, 95% CI:0.55–0.80)	[9]
Checkmate 057 (Phase III—NCT01673867)	EGFR mutant = 82 EGFR WT = 500	Nivolumab vs. Docetaxel	An EGFR-mutated subgroup analysis did not demonstrate PFS or OS benefit from nivolumab (HR = 1.18, 95% CI: 0.69–2.00)	[8]
OAK (Phase III—NCT02008227)	EGFR mutant = 85 EGFR WT = 628	Atezolizumab vs. Docetaxel	No OS benefit was shown with atezolizumab over docetaxel in NSCLC patients with EGFR mutations (HR = 1.24 95% CI:0.71–2.18) vs. WT (HR = 0.69, 95% CI:0.57–0.83)	[10]
Pooled Analysis (Checkmate 057, KEYNOTE 010, and POPLAR)	EGFR mutant = 186 EGFR WT = 1362	Pembrolizumab/Nivolumab/Atezolizumab vs. Docetaxel	ICIs did not improve OS vs. docetaxel in NSCLC patients with EGFR mutations (HR = 1.05 95% CI:0.70–1.55, *p* < 0.81) vs. WT (HR = 0.66, 95% CI:0.58–0.76, *p* < 0.0001)	[97]
Pooled Analysis (Checkmate 017, 057, 010, OAK, and POPLAR)	EGFR mutant = 271 EGFR WT = 1990	Pembrolizumab/Nivolumab/Atezolizumab vs. Docetaxel	ICIs did not improve OS vs. docetaxel in NSCLC patients with EGFR mutations (HR = 1.11 95% CI:0.80–1.53, *p* = 0.54) vs. WT (HR = 0.67, 95% CI:0.60–0.75, *p* < 0.001)	[106]

In contrast, few studies have also shown a favourable response with the combination of *EGFR*-TKIs with PD-1/PD-L1 pathway blockade therapy. CheckMate-012 evaluated nivolumab in combination with erlotinib (n = 20) in advanced NSCLC patients with acquired resistance to *EGFR* TKI. Four out of twenty patients showed a favourable response with the addition of nivolumab and an ORR of 15%, including one complete response [107]. Another trial evaluated the safety and tolerability of atezolizumab plus erlotinib in NSCLC patients previously untreated or treated with one prior non-*EGFR* TKI therapy (n = 28). The combination of erlotinib and atezolizumab showed a tolerable safety profile and favourable efficacy compared with previous reports of erlotinib monotherapy, with a median PFS of 15.4 months and ORR of 75% [108].

Currently, the guidelines by NCCN do not recommend immune checkpoint inhibitors for patients with *EGFR* mutations (https://www.nccn.org/patients/guidelines/content/PDF/lung-metastatic-patient.pdf (accessed on 19 January 2025)). Additionally, these combinations pose a severe toxicity risk to patients and a major challenge. The underlying mechanisms mediating the lower clinical efficacy of PD-1/PD-L1 blockade therapy are unclear. Further investigations into these mechanisms are warranted to specify patient populations that can benefit from the combination of treatments.

### 2.4. Tumour Mutation Burden (TMB)

Multistep carcinogenesis is driven by the accumulation of somatic mutations. These mutations can originate from either extrinsic factors, such as smoking, UV rays, radiation, etc., or from cell intrinsic factors, such as DNA damage repair defects [109]. TMB is generally defined as the number of acquired somatic mutations per megabase within a genomic sequence and is assessed by whole-exome sequencing (WES) or targeted panel next-generation sequencing (NGS). Mutations in genomic DNA may lead to the production of impaired/altered proteins. These altered proteins may serve as neoantigens; they can bind to the major histocompatibility complexes I and II (MHC I/II) in the endoplasmic reticulum and are then transported to the plasma membrane, where they are presented on the cell surface. The presentation of normal cellular peptides will not elicit an immune response due to central tolerance; however, if this peptide originates from a foreign organism or a mutated protein, it may be recognized by CD4^+^ and CD8^+^ T cells, as well as B cells, and trigger an immune response [110,111].

As neoantigens are a product of genomic mutations in tumour cells, TMB-High (TMB-H) tumours will be more immunogenic, as compared to TMB-Low (TMB-L) tumours, and will putatively show a better response to immune checkpoint inhibitors [112]. However, recent studies show that not all mutations have similar immunogenic effects. Mutations affecting RNA splicing, insertion/deletion mutations, and frameshift mutations have been shown to produce more immunogenic antigens as compared to the nonsynonymous single-nucleotide mutations [113,114]. Nevertheless, regardless of the mutation type and their variable contributions to tumour immunogenicity, the TMB scoring system weighs them equally and highlights the limitations of using TMB as a predictive biomarker for ICI treatment. The significance of TMB as a predictive biomarker and its association with patients’ responses has been thoroughly explored in multiple reviews [115,116].

### 2.5. Microsatellite Instability (MSI)

Mismatch repair (MMR) is a highly conserved DNA repair pathways that play a role in maintaining genomic integrity and DNA replication fidelity [117]. Genes involved in the MMR pathway include MLH1, MSH2, MSH6, and PMS2 [118]. MMR maintains genomic stability by correcting errors generated during DNA replication, including slippage and base substitutions, and prevents deletions and insertions of abnormal DNA at microsatellites. Microsatellites (MSs) are repeat sequences of 1–6 nucleotides, also known as Short Tandem Repeats (STRs). Multiple MSs can be found in chromosomes in a repeating pattern and are mostly located close to coding regions, but they can also be found in non-coding regions and introns. These repeat sequences are especially susceptible to errors if the mismatch repair pathway is impaired, e.g., by mutation or epigenetic inactivation. Cells with a dysfunctional MMR pathway (dMMR) cannot repair the errors that occur during DNA replication, creating an inconsistent number of MS repeats and resulting in microsatellite instability (MSI), a hallmark of MMR deficiency. If only one repeat is altered, the tumour is identified as MSI-Low (MSI-L), whereas if two or more mutated sequences are found, the tumour is considered to be MSI-High (MSI-H). Without any altered repeats, the tumour is known to have microsatellite stability (MSS). dMMR and MSI-H are often used interchangeably because of their high level of consistency among many tumours. dMMR/MSI-H have been widely detected in multiple tumours, including colorectal cancer [119], endometrial [120] and breast cancers [121], gastrointestinal adenocarcinoma [122], and lung adenocarcinoma [123].

Preclinical studies have shown that tumours with the MSI-H phenotype are hypermutated and express abnormal peptides that can become neoantigens and induce an immune response [124,125]. MSI-H/dMMR tumours are also associated with an increased expression of checkpoint proteins, such as PD-L1 and PD-1 [124,126,127,128,129]. Llosa et al. [130] analyzed the TME of colorectal cancer using flow cytometry, PCR, IHC, and functional TIL analysis. They showed that the dMMR subset of tumours selectively upregulated multiple checkpoint proteins, including PD-L1, PD-1, CTLA-4, IDO, and LAG3. This increased expression of both PD-1 and PD-L1 made these tumours more responsive to checkpoint inhibitors by inhibiting the PD-1/PD-L1 interaction and improving the ability of host immune cells to kill the tumour cells. Pembrolizumab was approved by the FDA for pediatric and adult patients with dMMR/MSI-H solid tumours [131,132,133]. This was the first time the FDA approved a treatment based on a genomic biomarker rather than tumour histology and site of origin. However, dMMR/MSI in lung cancer is uncommon and found only in 1.6% of lung cancers [134].

## 3. Non-Tumour-Associated Factors

### 3.1. Hypoxia

Hypoxia is one of the characteristics of the tumour microenvironment, where the oxygen supply to the tumour is either limited or there is an overconsumption of O_2_. Hypoxia is commonly observed in solid tumours because of the irregular tumour vasculature and has important implications in tumour progression [135]. Hypoxic conditions lead to the activation and stabilization of hypoxia-inducible transcription factors (HIFs). The main transcription factor that is activated by reduced O_2_ conditions is HIF-1, which is composed of two subunits: HIF-1α and HIF-1β. In normoxic cellular conditions, HIF-1α is hydroxylated on proline residues and is targeted for proteasomal degradation by the E3 ubiquitin ligase complex. In hypoxic conditions, enzymes responsible for hydroxylating HIF-1α are suppressed, leading to its stabilization. HIF-1α then translocates to the nucleus, where it forms a heterodimer with HIF-1β. This complex of HIF-1α/HIF-1β binds to hypoxia response elements (HREs), which results in the transcriptional upregulation of its target genes [136,137].

Hypoxia has immunosuppressive effects within the tumour microenvironment and affects the expression of immune checkpoint molecules by activating HIF-1α, regulating glycolysis and the Ado-A2a receptor (A2aR) axis, and driving EMT [138,139]. The role of hypoxia-associated molecules in up- and downregulating the expression of PD-L1 emphasizes its importance as a mechanism by which malignant tumours evade the immune system (Table 4). Multiple studies have shown that HIF-1α is a transcriptional modulator of PD-L [140,141,142]. Under hypoxic conditions, HIF-1α binds to hypoxia response elements on the PD-L1 promoter, including HRE2 and HRE4, and upregulates PD-L1 expression in cancerous and immune cells, including myeloid-derived suppressor cells, DCs, and macrophages [143,144]. A significant characteristic of a hypoxic tumour microenvironment is the accumulation of adenosine (Ado) in the stroma, which, in combination with A2aR and A2bR on the surface of CD8^+^ T cells, leads to an increased expression of cyclic adenosine monophosphate (cAMP) in CD8^+^ T cells. This accumulation of Ado eventually leads to PD-1 upregulation on CD8^+^ T cells and the inhibition of IFN secretion, hence mediating the functional inhibition of CD8^+^ T cells [145].

Tumour hypoxia is also a potent inducer of autophagy through multiple interlinked mechanisms. Under low-oxygen conditions, HIF-1α stabilization drives the transcription of genes such as BNIP3 and BNIP3L, which disrupt the Bcl-2–Beclin-1 complex and free Beclin-1 to initiate autophagosome formation [146]. Concurrently, energy stress from hypoxia activates AMPK and inhibits mTORC1, further promoting autophagy induction [147]. These pathways act together to enhance autophagic flux in hypoxic tumour regions. Autophagy can both enhance tumour immunogenicity and facilitate immune evasion, depending on context and cargo specificity. On the one hand, autophagy in dying cancer cells promotes immunogenic cell death (ICD) by enabling the release of damage-associated molecular patterns (DAMPs) and ATP, which recruit and activate dendritic cells (DCs) and cause T cell priming and, therefore, anti-tumour immunity [148]. Conversely, autophagy can promote immune evasion by selectively degrading key immune-recognition molecules, as it has been shown to target MHC class I molecules for lysosomal degradation, reducing surface MHC-I and impairing CD8^+^ T cell recognition [149]. Therefore, hypoxia-induced autophagy can exert a dualistic influence on the tumour immune microenvironment.

Low infiltration and exhaustion of CD8^+^ T cells is associated with anti-PD-1/PD-L1 resistance [150,151], and studies have shown that a hypoxic tumour microenvironment contributes to the exhaustion of T cells [152]. In mouse models, the infiltration of both CD4^+^ and CD8^+^ T cells is seen mainly in normoxic zones of the tumour, while the hypoxic zones lack tumour-infiltrating T cells [153]. The combination of TH-302, a hypoxia-activated prodrug that reverses hypoxia, with PD-1 and CTLA-4 inhibitors may improve survival in mouse tumours as compared to TH-302 alone or dual antibody alone. Reduced hypoxia also restored T cell infiltration and proliferation at the tumour site, allowing them to escape from the blood vessels into the tumour microenvironment [153]. Building on these findings, a phase I trial evaluated evofosfamide (TH-302) with ipilimumab in advanced solid malignancies. The combination was generally well tolerated and produced preliminary signs of activity, with an overall response rate of 17% and disease control in 83% of pretreated participants [154]. These findings suggest that hypoxia-targeting agents may recondition the tumour microenvironment and enhance the efficacy of immune checkpoint inhibitors, warranting further investigation.

**Table 4 cancers-17-02199-t004:** Role of hypoxia-associated molecules in modulating the expression of PD-1 and PD-L1.

Hypoxia-Associated Molecules	Effect on PD-1/PD-L1 Expression	Reference
HIF-1α	PD-L1—Stimulatory	[142,143,144,155]
CA9	PD-L1—Stimulatory	[156]
PKM2	PD-L1—Stimulatory	[157]
Lactate, GPR81, TEAD, TAZ	PD-L1—Stimulatory	[158]
LDH-A	PD-L1—Stimulatory	[159]
SNAI1	PD-L1—Stimulatory	[160]
ZEB1	PD-L1—Stimulatory	[160,161]
circPRDM4	PD-L1—Stimulatory	[142]
Ado, A2aR	PD-1—Stimulatory	[162,163]
miR-200	PD-L1—Inhibitory	[160,161]
cAMP, PKA	PD-L1—Inhibitory	[158]

### 3.2. Angiogenesis

Hypoxia induces a tumour to produce pro-angiogenic factors, such as VEGF, leading to the activation of angiogenic pathways [164] (Figure 3). Because of the aberrant hypersecretion of these factors, the maturation of these new blood vessels is impeded, leading to abnormal angiogenesis and leaky vessels [165]. The purpose of anti-angiogenic agents is to reduce the blood and oxygen supply to the tumour site and thus promote oxygen and nutrient starvation. VEGF and its receptor VEGF-R have been shown to be favourable targets for developing anti-angiogenic agents. Bevacizumab is an anti-VEGF monoclonal antibody and the first anti-angiogenic agent that was approved for multiple solid tumours, including metastatic non-small cell lung cancer [166].

For ICI efficacy, pre-existing TILs are crucial for anti-tumour effects. Firstly, aberrant angiogenesis reduces the number and function of anti-tumour lymphocytes through high interstitial fluid pressure, which makes it challenging for T cells to overcome the high pressure [167]. Secondly, at the same time, enhanced angiogenesis also increases the abundance of pro-tumour lymphocytes by secreting chemokines that recruit regulatory T cells and MDSCs to the tumour [168] and promote polarization of tumour-associated macrophages to the M2-like phenotype [169]. Thirdly, Fas ligand (Fas-L) on the tumour endothelial barrier causes the elimination of CD8^+^ T cells rather than regulatory T cells because of the high c-FLIP expression on regulatory T cells [170]. Therefore, because of the effects of hyper-angiogenesis on the immune makeup of the TME, especially in promoting pro-tumour immune populations, anti-angiogenic agents can be beneficial and supplement the anti-tumour effects of anti-PD-1/PD-L1 blockade.

VEGF signalling has been shown to directly enhance inhibitory checkpoint expression on both immune and tumour cells. In murine tumour models, elevated VEGF-A levels increased PD-1 on CD8^+^ T cells via VEGFR2 and downstream NFAT-dependent pathways, and blockade of VEGF-A/VEGFR reversed this upregulation and alleviated T cell exhaustion [171]. Similarly, VEGF, acting through VEGFR2 on myeloid cells, including MDSCs and tumour-associated macrophages, elevates PD-L1 levels, reinforcing their suppressive phenotype [172]. Beyond immune cells, VEGF co-receptors can sustain PD-L1 on tumour cells. VEGF/NRP2 signalling was reported to maintain PD-L1 expression in prostate cancer cells, and interrupting this axis lowered PD-L1 levels and improved T cell-mediated killing [173]. Collectively, these data demonstrate that VEGF-driven pathways directly upregulate PD-1 or PD-L1 across multiple cell types, providing a mechanistic rationale for combining anti-angiogenic agents with PD-1/PD-L1 blockade to counteract VEGF-mediated immunosuppression.

Interaction between angiogenesis and immunity can promote tumour immune escape and resistance to ICI blockade. There are currently 19 NSCLC active clinical trials that are evaluating bevacizumab, and 38 are in the recruiting phase (https://clinicaltrials.gov). Using a mathematical model, Schmidt et al. evaluated the synergistic effects of various treatments in combination with anti-PD-1/PD-L1, including chemotherapy, anti-CTLA-4, and anti-angiogenic agents. The study found that the combination of anti-PD-1 with anti-angiogenic agents provided the strongest synergistic effects when compared to other combination treatments [174]. Additionally, the Impower150 trial evaluated the effects of combination therapy, including bevacizumab, atezolizumab, and chemotherapy, in treatment-naïve NSCLC patients [175]. A total of 400 patients received atezolizumab plus bevacizumab plus carboplatin plus paclitaxel (ABCP group) and 400 patients received bevacizumab plus carboplatin plus paclitaxel (BCP group). The results showed that the ORR of the ABCP cohort was significantly higher than that of the BCP cohort (ORR: 63.5% vs. 48.0%, 95%CI: 58.2–68.5% vs. 42.5–53.6%). Similar results were found in OS and PFS analyses [175].

However, while anti-angiogenic agents inhibit tumour growth, they have also been reported to increase tumour invasiveness, distant metastasis, and eventual development of resistance to the treatment [176,177]. Their effects might also be transient, as these therapies can induce initial vessel normalization, but over time, their efficacy wanes as compensatory pro-angiogenic signalling accumulates [165]. These treatments also carry notable toxicity profiles that constrain dosing and duration [178].

### 3.3. CAF-Mediated TGF-β Signalling

Cancer-associated fibroblasts (CAFs) are a heterogeneous stromal cell population in the TME that contributes to extracellular matrix (ECM) remodelling and the secretion of immunomodulatory factors, such as TGF-β. CAFs deposit and reorganize collagen and other matrix proteins, creating physical barriers that impede T cell infiltration into tumour nests [179]. They also secrete chemokines and cytokines, including TGF-β, which recruit and support immunosuppressive cell populations while directly inhibiting cytotoxic T lymphocyte function. TGF-β signalling, often upregulated in CAFs and tumour cells, drives the EMT in cancer cells [180,181], upregulates PD-L1 expression [182,183], and promotes the differentiation or expansion of regulatory T cells [184], collectively dampening anti-tumour immunity and contributing to resistance to PD-1/PD-L1 blockade.

Several in vitro and in vivo studies have elucidated how CAF-derived TGF-β modulates immune populations in the TME. Tumours with increased levels of CAFs have been associated with high levels of Treg infiltration [185,186]. In co-culture experiments, CAFs produce TGF-β, which induces FOXP3 expression in naïve CD4^+^ T cells, promoting Treg differentiation and suppressing effector T cell responses [184]. TGF-β treatment of NSCLC cell lines also upregulates PD-L1 mRNA and protein expression on tumour cells via SMAD3 [182] and non-canonical pathways [183], further promoting immune escape. In murine tumour models, blocking TGF-β signalling in CAF-rich, immune-excluded tumours increases CD8^+^ T cell infiltration and restores responsiveness to PD-1/PD-L1 inhibitors. Notably, a study by Mariathasan et al. [187] demonstrated that stromal TGF-β activity contributes to the exclusion of T cells and attenuates the response to anti–PD-L1 therapy. In addition, combining TGF-β blockade with PD-L1 inhibition converted immune-excluded tumours into inflamed phenotypes with enhanced T cell infiltration and anti-tumour activity.

More recent preclinical work has explored the dual inhibition of PD-L1 and TGF-β. Combined PD-L1/TGF-β blockade in immune-excluded tumour models permits expansion and differentiation of stem cell-like CD8^+^ T cells, replacing dysfunctional-exhausted T cell pools and generating IFNg^hi^ effector cells that remodel myeloid and stromal niches into an immune-supportive TME [188]. These studies highlight that TGF-β signalling restrains intratumoural T cell function and effector differentiation and that dual inhibition can overcome barriers to anti-tumour immunity.

Several early-phase trials combining TGF-β pathway inhibitors with PD-1/PD-L1 blockade have reported modest but notable response rates in patients. Bintrafusp alfa (M7824), a fusion protein combining an anti–PD-L1 antibody with a TGF-β “trap”, has been evaluated in multiple cancer types. In a phase I study of bintrafusp alfa, the overall objective response rate (ORR) across 80 patients with advanced NSCLC was 21.3%. Notably, PD-L1-high tumours (≥80% expression) had an ORR of 85.7% at high dose levels [189]. In a phase Ib/II trial of the TGF-βRI inhibitor galunisertib plus nivolumab in solid tumours including NSCLC, 24% of patients achieved confirmed partial responses, and the median duration of response was 7.4 months [190]. These combination strategies demonstrate that dual blockade can elicit responses in subsets of patients, though efficacy varies, and further biomarker-driven studies are needed.

### 3.4. Immune Cell Heterogeneity

Only a subset of patients derive durable benefit from ICI treatment, underscoring the importance of understanding how heterogeneity among immune cells influences therapeutic outcomes. The functional state of T cells is integral in modulating the response to immunotherapies. In melanoma patients who progressed on anti-CTLA-4 therapy but responded to anti-PD-1, the infiltration of T cells and their cytolytic activity increased [191]. Clinical studies have revealed that a notable subset of tumour-infiltrating regulatory T cells express PD-1, and their abundance correlated with poorer outcomes following PD-1 blockade. For instance, in metastatic clear cell renal cell carcinoma, higher levels of PD-1^+^ intratumoural Tregs were associated with a lack of benefit from anti-PD-1 therapy, suggesting that these cells contribute to an immunosuppressive TME that resists PD-1 inhibition [192]. In addition to Tregs, MDSCs and TAMs have also been associated with poor responses to anti-PD-1/PD-L1 therapy, as these cells secrete immunosuppressive mediators, including *TGF-β* and IL-10 [193,194].

TGF-β-rich stromal niches, often orchestrated by CAFs, drive immune exclusion and hinder T cell infiltration of tumours in patients receiving PD-1/PD-L1 inhibitors. In colorectal cancer samples, high TGF-β signalling signatures in the stroma were linked to a lack of response to anti-PD-L1 therapy [195], and dual blockade of TGF-β and PD-1/PD-L1 in early clinical investigations showed promise in converting non-T cell-inflamed tumours into inflamed phenotypes [196]. Hence, TGF-β signalling or modulating CAF activity alongside PD-1/PD-L1 blockade may overcome stromal-mediated immune suppression.

Several studies have indicated that T cells can adapt to PD-1 blockade by increasing the expression of inhibitory receptors, thereby undermining monotherapy. For instance, in NSCLC with STK11/KEAP1 mutations resistant to PD-(L)1 inhibitors, adding CTLA-4 blockade may restore responses, implying compensatory CTLA-4 upregulation [45]. Similarly, the co-upregulation of PD-1 and LAG-3 on exhausted TILs has been observed in resistant tumours, and dual PD-1/LAG-3 inhibition can reinvigorate these cells [197]. Preclinical evidence also shows that TIGIT is often co-expressed with PD-1 on tumour-specific CD8^+^ T cells, and combined TIGIT/PD-1 blockade may enhance anti-tumour activity [198]. Hence, monitoring these changes in checkpoint expression could guide the selection and timing of combination regimens to overcome adaptive resistance.

### 3.5. Gut Microbiota

Increasing evidence shows the role of the microbiome not only in cancer development and progression but also in cancer treatments, resulting in its recognition as an emerging hallmark of cancer [199]. In addition to the intestinal microbiome, the local TME has been shown to constitute its own distinct microbiome, including lung cancer, where the airway microbiome has been associated with lung cancers [200]. The microbiota of the airway may induce the activation of γδ T cells, which produce interleukin-17 and cause chronic inflammation that plays a role in the development of lung adenocarcinoma [201].

The gut microbiota composition has been shown to impact the host immune system, and studies have now suggested that patients who respond to anti-PD-1 blockade have a different microbiota than non-responders. A meta-analysis showed no significant difference in the diversity of the microbiome between responders and non-responders, suggesting that specific microbial species may have driven the observed differences between the two groups [202]. The study also identified 17 microbial species that were differentially present in responders as compared to non-responders, but the study was limited to metastatic melanoma patients, and the results may not be generalizable to other cancer types [202]. NSCLC patients who responded to anti-PD-1 were shown to have increased levels of *Bifidobacterium bifidum* as compared to non-responders [203]. Additionally, Tomita et al. evaluated the effects of *Clostridium butyricum* (CBT), a commonly used probiotic in Japan, on anti-PD-1/PD-L1 therapeutic efficacy in 118 patients with advanced NSCLC. Survival analysis showed that the administration of probiotic CBT in addition to ICI significantly prolonged the OS and PFS in these patients [204].

Multiple studies have proposed the mechanism through which the gut microbiota may influence the response to immune checkpoint inhibitors by modulating the host immune system. First, antigen mimicry can affect the anti-tumour response, where some strains can express epitopes that are homologous to MHC tumour epitopes. These microbial strains can then induce and activate CD8^+^ T cells, which can then recognize tumour cells, eliminate them, and, consequently, reduce the tumour burden [205,206]. Second, microbial-associated molecular patterns (MAMPs) can engage pattern recognition receptors (PRRs), such as TLRs, that are expressed on innate immune cells [207]. Several bacterial species can produce CpG DNA motifs, which can bind TLR9 and induce the secretion of pro-inflammatory cytokines, such as type I interferons, by myeloid cells [208]. Third, metabolites derived from the microbiome have been known to modulate immune functions and potentially influence the response to ICI treatment. Short-chain fatty acids produced by the microbiota, including acetate, propionate, and butyrate, have been hypothesized to impact ICI efficacy [209]. For example, high concentrations of propionate and butyrate in fecal matter have both been significantly associated with long-term survival to anti-PD-1 treatment in NSCLC patients [210]. Among these short-chain fatty acids, butyrate has been shown to induce TGF-β expression and aid in Treg differentiation by promoting transcription of Foxp3 [211,212], indicating that TGF-β levels in the gut can impact the function of immune populations. And finally, specific bacterial species can be indicative of the response to immunotherapy. Patients with high levels of *Faecalibacterium*, Ruminococcaceae, and *Clostridiales* showed increased systemic CD4^+^ and CD8^+^ T cells in the circulation as well as a higher cytokine response to a PD-1 inhibitor, whereas patients with high *Clostridiales* showed a higher proportion of regulatory T cells and MDSCs, as well as a decreased cytokine response to anti-PD-1. Additionally, microbial products can escape through the blood to the lymph nodes, enhance the cytokine response peripherally, and contribute to favourable immune stimulation [213].

A phase I trial combining a healthy-donor fecal microbial transplant (FMT) with pembrolizumab or nivolumab in 20 previously untreated advanced melanoma patients showed an ORR of 65% and a complete response in 20% of the patients. Microbiome profiling showed an increase in microbiome similarity in donors and patients in the responders [214]. Additionally, two investigator-initiated phase I/II trials in NSCLC [215] and melanoma [216] are also underway, testing FMT with ICIs. Continued biomarker analyses (e.g., microbial taxa dynamics and immune profiling) from these trials will be critical in shedding light on which patients are most likely to benefit from such therapies.

## 4. Conclusions and Future Directions

PD-1/PD-L1 inhibitors have had a profound impact on the management and treatment of NSCLC patients. This has reinvigorated the interest of researchers and oncologists in the tumour microenvironment, which was often overlooked in the past. High PD-L1 expression, high tumour mutation burden, and microsatellite instability are the most robust predictive biomarkers for the response to PD-1/PD-L1 checkpoint therapy. This review addresses both tumour-associated and non-tumour-associated biomarkers that may predict the response to PD-1/PD-L1 blockade in NSCLC patients (Figure 4). It is important to note that all the potential biomarkers mentioned above do not act in isolation to improve or impair the anti-tumour response, but instead, they work in tandem within the tumour immune microenvironment and even influence each other, often contributing to a pro-tumour or anti-tumour response. Personalizing treatment regimens based on the status of multiple biomarkers will provide further avenues to improve patient responses. Evidence from recent studies emphasizes the value of profiling *STK11* and *KEAP1* mutations to personalize ICI treatment. Future approaches should therefore consider combining STK11/KEAP1 mutation status, TMB, PD-L1 expression, and microbiome profiling to identify patients who are most likely to benefit from monotherapy vs. combination treatments and to guide interventions, such as microbiota modulation alongside immunotherapy.

Notably, high-dimensional analyses, including single-cell RNAseq, imaging mass cytometry, and spatial transcriptomics, have revealed spatial niches of T cells and macrophages that correlate with survival in lung tumours, suggesting potential biomarkers for immunotherapy response prediction [217]. In parallel, integrating spatial architecture mapping with genomic profiling in early-stage lung cancer has identified distinct TME archetypes associated with differential clinical outcomes and likely differing sensitivities to immunotherapy [218]. These studies exemplify how emerging spatial and multi-omics technologies may potentially improve patient stratification to guide the selection of immunotherapies.

However, more research is warranted to identify how a non-immunogenic TME can be transformed into an immunogenic TME. Both chemotherapeutic agents and radiation have been shown to improve tumour immunogenicity by increasing immune cell infiltration, DNA damage response, tumour mutation burden, upregulating PD-L1, and the cytokine response. Multiple clinical trials have shown favourable responses in NSCLC patients with ICI and chemotherapy combinations; however, further studies are required to understand the mechanisms of the enhanced response. In conclusion, further research is needed to refine the currently existing biomarkers and to develop new ones for evaluating the patient response.

## Figures and Tables

**Figure 1 cancers-17-02199-f001:**
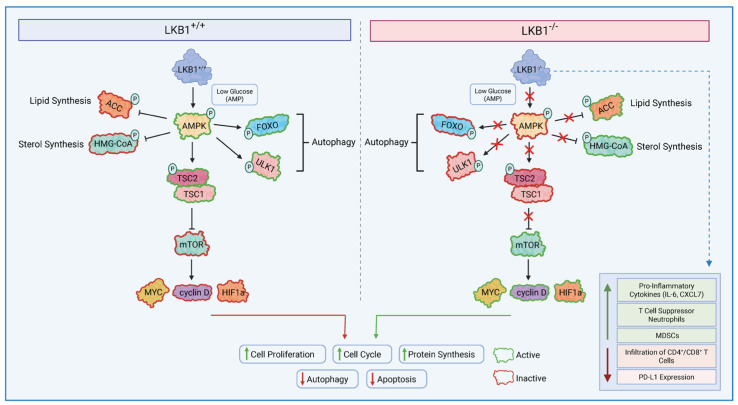
LKB1 Dependent Signalling. LKB1 directly phosphorylates and activates AMP-activated protein kinase in nutrient-deficient and hypoxic conditions; in turn, AMPK phosphorylates the TSC2 complex to mediate the effects on cell growth. AMPK activation, therefore, suppresses mTORC1-dependent transcriptional regulators, such as MYC, HIF1α, and cyclin D, which play a role in promoting cell growth and tumourigenesis. Conversely, when LKB1 is mutated, it does not lead to the successful activation of AMPK in low intracellular ATP conditions. This will prevent the activation of tumour suppressors TSC1/TSC2, leading to constant hyperactivation of the mTORC1 complex. mTORC1 then promotes tumourigenesis via the transcriptional activation of MYC, HIF1α, and cyclin D. LKB1 alterations also modulate the tumour immune microenvironment by increasing immune suppressive subsets and reducing the infiltration of CD4^+^/CD8^+^ T cells. LKB1: liver kinase B1, TSC1/1: tuberous sclerosis complex 1/2, mTORC1: Mammalian Target of Rapamycin Complex 1, HIF1α: Hypoxia-Inducible Factor.

**Figure 2 cancers-17-02199-f002:**
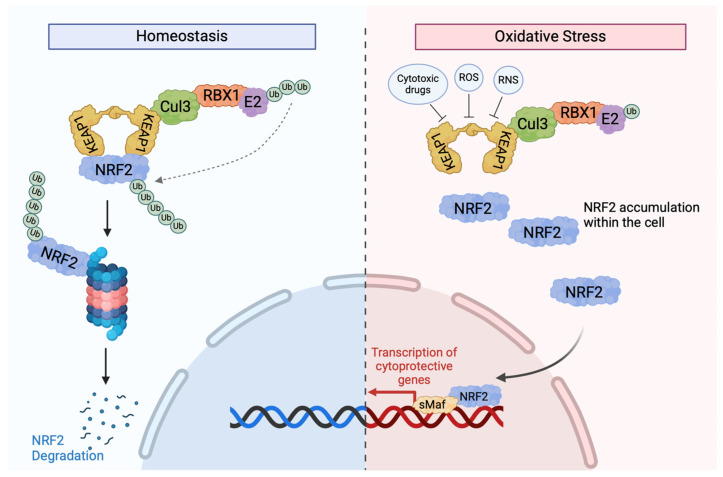
KEAP1-dependent signalling. In normal circumstances, NRF2 interacts with two KEAP1 molecules via the ETGE and DLG motifs in its Neh2 domain to cause NRF2 ubiquitination via the activation of the Cul3-based E3 ligase complex [51]. After its ubiquitination, NRF2 is marked for degradation by the 26S proteosome, with low cytoplasmic levels. When cells are exposed to oxidative stress or cytotoxic drugs, such as chemotherapies, sensor cysteines in KEAP1, especially Cy151, interact with electrophiles and ROS, causing conformational changes and resulting in the detachment of NRF2 from KEAP1 and the disruption of KEAP1-mediated NRF2 degradation [52]. This reduces the binding affinity between the two proteins, allowing NRF2 to translocate to the nucleus to activate the transcription of various protective genes. Once oxidative stress is removed, KEAP1 translocates to the nucleus and brings NRF2 out to the cytoplasmic Cul-E3 ubiquitin ligase complex for degradation [51].

**Figure 3 cancers-17-02199-f003:**
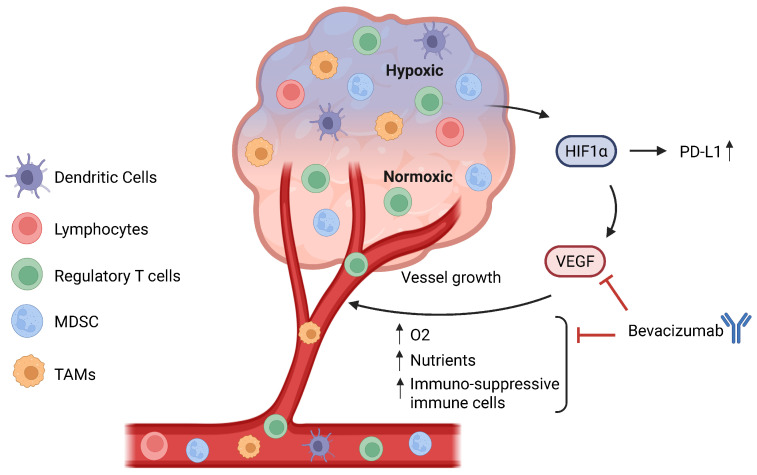
Relationship between hypoxia and angiogenesis. A hypoxic TME will lead to the activation and stabilization of hypoxia-inducible transcription factors (HIFs), especially HIF-1α, which directly binds to the PD-L1 promoter to upregulate its expression and also plays a role in angiogenesis. HIF-1α directly upregulates VEGF, leading to the formation of new blood vessels, which increase the oxygen and nutrient supply to the tumour. Additionally, angiogenesis also leads to greater infiltration of immune cells, especially regulatory T cells, MDSCs, and TAMs, hence contributing to the immunosuppressive TME. Bevacizumab, an anti-VEGF, can inhibit the binding of VEGF to its receptor VEGF-R and prevent the formation of new blood vessels and, by extension, an immunosuppressive TME. MDSCs: myeloid-derived suppressor cells. TAMs: tumour-associated macrophages. TME: tumour microenvironment.

**Figure 4 cancers-17-02199-f004:**
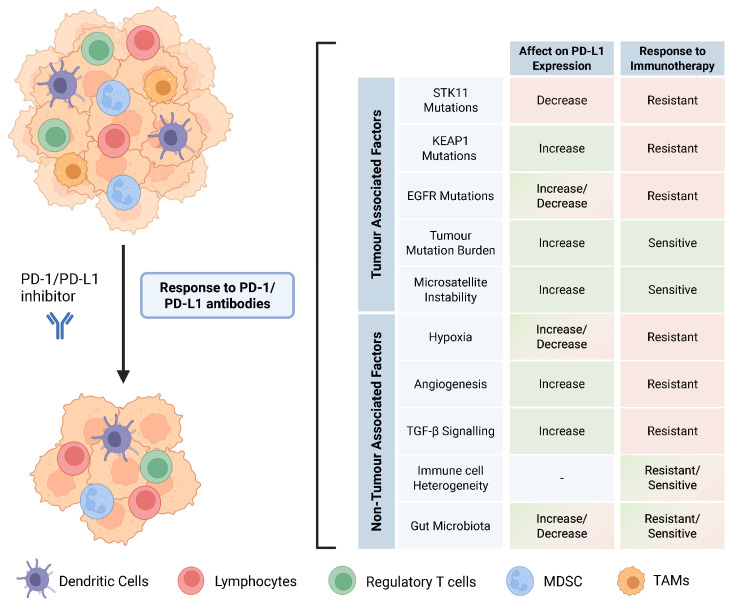
Tumour-associated and non-tumour-associated factors that confer response to ICIs in NSCLC.

**Table 1 cancers-17-02199-t001:** Approved PD-1 and PD-L1 inhibitors for NSCLC treatment.

Antibody	Target	Date of Approval	Company	Stage	Approval of Use
**Approved by FDA**					
Pembrolizumab	PD-1	2014	Merck, Rahway, NJ, USA	Stage IB, II, or IIIA	1 L, ≥2 L
Nivolumab	PD-1	2014	Bristol Myers, Princeton, NJ, USA	Stage IIA to IIIB	1 L, ≥2 L
Atezolizumab	PD-L1	2016	Genentech/Roche, Basel, Switzerland	Stage II to IIIA	1 L, Maintainenace
Durvalumab	PD-L1	2017	AstraZeneca, Cambridge, UK	Stage III	Post-chemoradiation
Cemiplimab	PD-1	2018	Regeneron, Tarrytown, NY, USA	Stage IV	1 L
**Approved in China**					
Sintilimab	PD-1	2018	Eli Lilly/Innovent Biologics, Suzhou, China	Stage IV	1 L, ≥2 L
Camrelizumab	PD-1	2019	Jiangsu Hengrui Pharma Co., Ltd. Lianyungang, China	Stage IV	1 L, ≥2 L
Penpulimab	PD-1	2021	Akeso Biopharma Co., Zhongshan, China	Stage III to Stage IV	1 L
Serplulimab	PD-1	2022	Shanghai Henlius Biotech, Inc., Shanghai, China	Stage III to Stage IV	1 L

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
