# Peer review of "Tumour- and Non-Tumour-Associated Factors That Modulate Response to PD-1/PD-L1 Inhibitors in Non-Small Cell Lung Cancer"

_cancers, 2025, doi:10.3390/cancers17132199_

Round 1

Reviewer 1 Report

Comments and Suggestions for Authors

The review is interesting and well written. I would advise publication after addressing some issues listed below:

  1. The authors say at some point: “However, it is generally recognized that PD-L1 expression is not the most accurate predictor of response to ICI due to tumour heterogeneity and the inducible nature of PD-L1.”. I would rather say that it is an imperfect predictor of response, since claiming that it is not the most accurate predictor seems to imply that there are others which are better.
  2. In the TMB section, the authors claim that neoantigens will be presented in MHC I. Let me remark that anti-tumor immune responses in immunotherapies also heavily rely on MHC II presentation, and many neoantigens will be cross-presented to CD4 T cells. Indeed, CD4 T cell responses are critical as well for the efficacy of immunotherapies including in lung cancer. This should be mentioned as well. Some papers describing this are: PMID: 31645760, PMID: 31871119, PMID: 39349759
  3. I have missed in the review a section on some well-known mechanisms such as upregulation of additional immune checkpoint molecules in T-cells that bypass PD-1/PD-L1 blockade. For example, the main ones such as CTLA4 and LAG3 upregulation, but also others such as TIGIT, GITR, TIM3. For example see just a sample of papers on this: PMID: 39385035, PMID: 37291608, PMID: 37620318, PMID: 39030301.

My advise would be to include a section on this, which would improve their review.

Author Response

All changes and additions are highlighted by red font in the revised manuscript

Comment 1: The authors say at some point: “However, it is generally recognized that PD-L1 expression is not the most accurate predictor of response to ICI due to tumour heterogeneity and the inducible nature of PD-L1.”. I would rather say that it is an imperfect predictor of response, since claiming that it is not the most accurate predictor seems to imply that there are others which are better.

Response: This has now been revised as suggested, page 2, line 64.

Comment 2: In the TMB section, the authors claim that neoantigens will be presented in MHC I. Let me remark that anti-tumor immune responses in immunotherapies also heavily rely on MHC II presentation, and many neoantigens will be cross-presented to CD4 T cells. Indeed, CD4 T cell responses are critical as well for the efficacy of immunotherapies including in lung cancer. This should be mentioned as well. Some papers describing this are: PMID: 31645760, PMID: 31871119, PMID: 39349759

Response: This has now been modified. Page 11, Lines 405 & 409; PMID: 31645760 has been added to the reference list (reference 111).

 Comment 3: I have missed in the review a section on some well-known mechanisms such as upregulation of additional immune checkpoint molecules in T-cells that bypass PD-1/PD-L1 blockade. For example, the main ones such as CTLA4 and LAG3 upregulation, but also others such as TIGIT, GITR, TIM3. For example, see just a sample of papers on this: PMID: 39385035, PMID: 37291608, PMID: 37620318, PMID: 39030301. My advice would be to include a section on this, which would improve their review.

Response: We thank the reviewer for this suggestion. This has been addressed in a new section 3.4, page 16, Lines 647-656.

Reviewer 2 Report

Comments and Suggestions for Authors

1] In Figure 1, consider adding more detail on the downstream effectors of AMPK beyond mTORC1 to provide a more comprehensive picture of the pathway's complexity.

2] For scientific completeness, consider adding quantitative data on the frequency of key mutations (STK11, KEAP1) across different NSCLC subtypes to provide better context.

3] The section on LKB1 mutations mentions several cytokines (IL-6, CXCL7, CCL5, CXCL10). Consider adding a figure illustrating the cytokine signaling network in the tumor microenvironment to enhance understanding of these complex interactions.

4] For the studies cited regarding LKB1 mutations (lines 139-146), consider including a more critical assessment of their limitations.

5] Consider adding a comprehensive figure illustrating the complex interplay between different tumor and non-tumor factors that affect response to PD-1/PD-L1 inhibitors, which would enhance understanding of the multifactorial nature of immunotherapy response.

6] Introduction covers the basics of PD-1/PD-L1 inhibition, consider enhancing the historical context of how these inhibitors were developed and first tested in NSCLC to provide better scientific grounding.

7] In section "2.2. KEAP1 mutations" consider expanding on how KEAP1 alterations specifically affect response to immunotherapy through NRF2 pathway activation and subsequent effects on the tumor microenvironment.

8] Consider adding a dedicated section on emerging biomarkers and technologies (e.g., spatial transcriptomics, multi-omics approaches) that may improve prediction of response to immunotherapy.

9] In section "2.1 STK11/LKB1 mutations" provide details on how LKB1 mutations specifically impact T-cell function beyond just describing lower infiltration.

Author Response

All changes and additions are highlighted by red font in the revised manuscript

Comment 1: In Figure 1, consider adding more detail on the downstream effectors of AMPK beyond mTORC1 to provide a more comprehensive picture of the pathway's complexity.

Response: Figure 1 has been modified to include additional effectors of AMPK along with their functions.

Comment 2: For scientific completeness, consider adding quantitative data on the frequency of key mutations (STK11, KEAP1) across different NSCLC subtypes to provide better context.

Response: The frequency of STK11 mutations in NSCLC subtypes have been included (Page 2, Line 75). For KEAP1 mutations this information was already included in section 2.2 (KEAP1 Mutations, page 6, line 223)

 Comment 3: The section on LKB1 mutations mentions several cytokines (IL-6, CXCL7, CCL5, CXCL10). Consider adding a figure illustrating the cytokine signaling network in the tumor microenvironment to enhance understanding of these complex interactions.

Response: Figure 1 has been modified to include the effect of LKB1 on TME.

Comment 4: For the studies cited regarding LKB1 mutations (lines 139-146), consider including a more critical assessment of their limitations.

Response: Limitations of the studies have now been included (Page 4, line 161).  

Comment 5: Consider adding a comprehensive figure illustrating the complex interplay between different tumor and non-tumor factors that affect response to PD-1/PD-L1 inhibitors, which would enhance understanding of the multifactorial nature of immunotherapy response.

Response: We appreciate the suggestion and have attempted to illustrate these complex relationships in Figure 4.

Comment 6: Introduction covers the basics of PD-1/PD-L1 inhibition, consider enhancing the historical context of how these inhibitors were developed and first tested in NSCLC to provide better scientific grounding.

Response: Thank you for the suggestion. We have now added this in the introduction, page 1 and 2, lines 36, 48-51.

Comment 7: In section "2.2. KEAP1 mutations" consider expanding on how KEAP1 alterations specifically affect response to immunotherapy through NRF2 pathway activation and subsequent effects on the tumor microenvironment.

Response: This has been included in section 2.2, page 6, line 229-242.

Comment 8: Consider adding a dedicated section on emerging biomarkers and technologies (e.g., spatial transcriptomics, multi-omics approaches) that may improve prediction of response to immunotherapy.

Response: This has been included in the future directions, page 18, Line 731-739.

Comment 9: In section "2.1 STK11/LKB1 mutations" provide details on how LKB1 mutations specifically impact T-cell function beyond just describing lower infiltration.

Response: Section 2.1 has been modified to include this (Page 4, lines 123-129).

Reviewer 3 Report

Comments and Suggestions for Authors

The manuscript provides a comprehensive review of tumor associated and non tumor factors influencing PD-1/PD-L1 inhibitor responses in NSCLC but lacks critical depth in several key areas related to the tumor microenvironment and associated pathways.

While the role of STK11/LKB1 and KEAP1 mutations in shaping an immunosuppressive TME is discussed, there is minimal exploration of cancer-associated fibroblasts which are pivotal in extracellular matrix remodeling, immune exclusion, and secretion of immunosuppressive cytokines like TGF-β. The omission of TGF-β signaling is a significant gap, as it directly regulates PD-L1 expression, promotes Treg differentiation, and drives epithelial-mesenchymal transition all of which contribute to immunotherapy resistance.

The manuscript briefly mentions hypoxia-induced PD-L1 upregulation but fails to link hypoxia to autophagy, a process that supports tumor survival under metabolic stress and modulates antigen presentation. Autophagy’s dual role in either enhancing immunogenicity or promoting immune evasion via degradation of MHC-I or PD-L1 recycling is not addressed, nor is its interplay with apoptosis key mechanisms that determine tumor immune interactions.

While immune cell infiltration is analyzed, the review overlooks the functional heterogeneity of immune subsets, such as exhausted versus effector T cells, and the immunosuppressive role of myeloid-derived suppressor cells  or TAM polarization, which are influenced by TGF-β and CAF-secreted factors.

The gut microbiota section could be strengthened by discussing microbial modulation of TGF-β or autophagy pathways, which are emerging as cross regulators of systemic immunity.

The conclusion does not propose therapeutic strategies targeting autophagy, CAFs, or TGF-β to reprogram the TME, despite preclinical evidence supporting their combinatorial potential with PD-1/PD-L1 blockade.

Author Response

All changes and additions are highlighted by red font in the revised manuscript.

Comment 1: The manuscript provides a comprehensive review of tumor associated and non tumor factors influencing PD-1/PD-L1 inhibitor responses in NSCLC but lacks critical depth in several key areas related to the tumor microenvironment and associated pathways.

Response: We thank the reviewer for the suggestion and have attempted to address these concerns by adding two new sections, one on “CAF mediated TGF-B signaling” (Section 3.3, page 15) and second on “immune cell heterogeneity” (Section 3.4, page 16). 

Comment 2: While the role of STK11/LKB1 and KEAP1 mutations in shaping an immunosuppressive TME is discussed, there is minimal exploration of cancer-associated fibroblasts which are pivotal in extracellular matrix remodeling, immune exclusion, and secretion of immunosuppressive cytokines like TGF-β. The omission of TGF-β signaling is a significant gap, as it directly regulates PD-L1 expression, promotes Treg differentiation, and drives epithelial-mesenchymal transition all of which contribute to immunotherapy resistance.

Response: A section on TGFB’s role in modulating TME has been added (Section 3.3).

Comment 3: The manuscript briefly mentions hypoxia-induced PD-L1 upregulation but fails to link hypoxia to autophagy, a process that supports tumor survival under metabolic stress and modulates antigen presentation. Autophagy’s dual role in either enhancing immunogenicity or promoting immune evasion via degradation of MHC-I or PD-L1 recycling is not addressed, nor is its interplay with apoptosis key mechanisms that determine tumor immune interactions.

Response: Section 3.2 has been modified to address this concern. Pages 12-13, Line 486-500.

Comment 4: While immune cell infiltration is analyzed, the review overlooks the functional heterogeneity of immune subsets, such as exhausted versus effector T cells, and the immunosuppressive role of myeloid-derived suppressor cells or TAM polarization, which are influenced by TGF-β and CAF-secreted factors.

Response: This has been included in the new section 3.4 (page 16)

Comment 5: The gut microbiota section could be strengthened by discussing microbial modulation of TGF-β or autophagy pathways, which are emerging as cross regulators of systemic immunity.

Response: A comment on this has been added to section 3.5 (originally 3.3), page 17, Line 694-697.

Comment 6: The conclusion does not propose therapeutic strategies targeting autophagy, CAFs, or TGF-β to reprogram the TME, despite preclinical evidence supporting their combinatorial potential with PD-1/PD-L1 blockade.

Response: This comment has been addressed in section 3.3, page 15 (TGF-B signalling)

Reviewer 4 Report

Comments and Suggestions for Authors

Overall Assessment

The manuscript is promising, with clear writing, logical subsections, and supportive figures/tables. However, it requires comprehensive revisions to address gaps in recent literature, therapeutic strategies, mechanistic insights, and integration of tumor and non-tumor factors.

Comments

Abstract:

For a review article, the abstract should summarize the scope (tumor and non-tumor factors), key findings (e.g., specific factors like STK11, KEAP1, EGFR mutations), and clinical implications. If available, please share the abstract for a more precise evaluation.

Introduction:

- More recent references (2022–2025) should be used to reflect the latest advances in NSCLC immunotherapy, ICI resistance, novel biomarkers, or non-tumor factors.

- Non-tumor-associated factors not detailed in the provided manuscript that modulate ICI response.

- Tumor and non-tumor factors interaction with PD-L1 expression is missed.

- Tregs data regarding PD-1 expression is not integrated.

- Some recent therapeutic strategies regarding PD1-PDL1 ligation (such as CAR T-cells and CAR-NK cells) are not noted.

- Update Table 1 with any new ICI approvals (2023–2025):

Potential Addition: Tislelizumab (Tevimbra):

New Approval (March 2024): FDA approved tislelizumab for unresectable or metastatic esophageal squamous cell carcinoma (ESCC) post-chemotherapy. While not NSCLC-specific, tislelizumab is a PD-1 inhibitor, and ongoing trials (e.g., phase 3 for NSCLC) suggest future relevance.

Tumor-Associated Factors:

- Lacks discussion of potential therapeutic strategies to overcome LKB1-mediated ICI resistance (such as combination therapies).

- Add a sentence on potential strategies to target LKB1-mutated tumors (such as mTOR inhibitors, STING agonists).

- Limited discussion of KEAP1/NRF2 interactions with other mutations (such as KRAS, STK11).

- Expand on potential NRF2-mediated resistance mechanisms (such as specific drug-metabolizing enzymes).

- Discuss briefly KEAP1 co-mutations (such as with STK11) to align with the review’s comprehensive scope.

- Discuss briefly reasons for PD-L1 expression variability (such as different assays and cutoffs).

- Add mechanistic insights into ICI resistance (such as EGFR-driven immune suppression pathways).

- The manuscript mentions tumor mutational burden (TMB) and microsatellite instability (MSI) in references [86–108] but does not discuss them in the provided text. These are critical factors for ICI response and should be included in the review.

Non-Tumor-Associated Factors:

- Lacks NSCLC trial data.

- Discuss hypoxia measurement methods (such as PET imaging, HIF-1α staining) to strengthen translational relevance.

- Limited mechanistic depth on how VEGF signaling directly modulates PD-1/PD-L1 expression.

- Does not discuss resistance to anti-angiogenic therapies or their limitations (such as toxicity and transient effects).

- Limited discussion of overcoming resistance (such as anti-angiogenic failures and microbiome modulation challenges).

- Lacks recent studies for all sections.

Conclusions:

Highlight clinical implications (such as personalized ICI use based on mutation status).

Propose specific future directions (such as combining STK11/KEAP1 testing with TMB, microbiome modulation).

Author Response

All changes and additions are highlighted by red font in the revised manuscript

Abstract:

Comment 1: For a review article, the abstract should summarize the scope (tumor and non-tumor factors), key findings (e.g., specific factors like STK11, KEAP1, EGFR mutations), and clinical implications. If available, please share the abstract for a more precise evaluation.

Response: Abstract of the paper is provided on page 1 along with a simple summary.

Introduction:

Comment 2: More recent references (2022–2025) should be used to reflect the latest advances in NSCLC immunotherapy, ICI resistance, novel biomarkers, or non-tumor factors.

Response: Recent references have been added. Ref 13-15, 19-20.

Comment 3: Non-tumor-associated factors not detailed in the provided manuscript that modulate ICI response.

Response: We have added two new sections to address these concerns, section 3.3 (page 15) on TGF-B signaling and section 3.4 (page 16) on immune cell heterogeneity. 

Comment 4: Tumor and non-tumor factors interaction with PD-L1 expression is missed.

Response: These have been highlighted in sections 2.1 (Page 4, line 137-152), section 2.2 (page 7, line 253-262), section 2.3 (page 8-9, line 308-317), section 2.5 (page 12, line 445-452), section 3.1 (page 12, line 473-480, table 4), section 3.2 (page 14, line 539-551), and section 3.3 (page 15, line 599-603).

Comment 5: Tregs data regarding PD-1 expression is not integrated.

Response: This has been highlighted in section 3.4 (page 16).

Comment 6: Some recent therapeutic strategies regarding PD1-PDL1 ligation (such as CAR T-cells and CAR-NK cells) are not noted.

Response: We thank the reviewer for this suggestion. However, we feel a discussion on CAR-T cells is outside the scope of this review, as itself will need to be quite extensive.   

Comment 7: Update Table 1 with any new ICI approvals (2023–2025):

Potential Addition: Tislelizumab (Tevimbra):

New Approval (March 2024): FDA approved tislelizumab for unresectable or metastatic esophageal squamous cell carcinoma (ESCC) post-chemotherapy. While not NSCLC-specific, tislelizumab is a PD-1 inhibitor, and ongoing trials (e.g., phase 3 for NSCLC) suggest future relevance.

Response: We appreciate this suggestion. However, Table 1 focusses solely on the approved anti-PD-1/PD-L1 agents in NSCLC. By adding one non-NSCLC approved inhibitor, others will also need to be included, which distracts the intention of this Table. 

Tumor-Associated Factors:

Comment 8: Lacks discussion of potential therapeutic strategies to overcome LKB1-mediated ICI resistance (such as combination therapies).

Response: The section has now been modified to include this. Page 4, Lines 163-174.  

Comment 9: Add a sentence on potential strategies to target LKB1-mutated tumors (such as mTOR inhibitors, STING agonists).

Response: STING activation as a potential therapeutic strategy is now included (Page 4, Lines 169-172) because it modulates the tumour immune microenvironment, however since mTOR inhibitors have shown to cause immune suppression, these were not included.

Comment 10: Limited discussion of KEAP1/NRF2 interactions with other mutations (such as KRAS, STK11).

Response: Comment #8 and 10 are now addressed in section 2.2. Page 7, Lines 271-285.

Comment 11: Expand on potential NRF2-mediated resistance mechanisms (such as specific drug-metabolizing enzymes).

Response: Paragraph 2 in section 2.2 highlights this (page 5, line 195-207).

Comment 12: Discuss briefly KEAP1 co-mutations (such as with STK11) to align with the review’s comprehensive scope.

Response: A brief discission on KEAP1 co-mutations with STK11 and KRAS have been included in section 2.2. Page 7, Lines 271-285)

Comment 13: Discuss briefly reasons for PD-L1 expression variability (such as different assays and cutoffs).

Response: We thank the reviewer for this request. However, we feel this topic is outside the scope of this paper’s context, as it is extensive and has been the topic of several review manuscripts that one of us (M.-S. Tsao) was involved in (PMIDs: 29800747, 29053400, 31383961, 33580222, 36184066)

Comment 14: Add mechanistic insights into ICI resistance (such as EGFR-driven immune suppression pathways).

Response: Section 2.3 has been modified to include these suggestions. Page 9, Line 318-325.

Comment 15: The manuscript mentions tumor mutational burden (TMB) and microsatellite instability (MSI) in references [86–108] but does not discuss them in the provided text. These are critical factors for ICI response and should be included in the review.

Response: Sections 2.5 and 2.6 discuss both of these topics, the reason TMB is not explored in detail in section 2.5 is because there are multiple reviews that thoroughly discuss the topic in great detail.

Non-Tumor-Associated Factors:

Comment 16: Lacks NSCLC trial data.

Response: Sections 3.1 (Page 10, line 509-515), 3.3 (Page 15, Line 615-625), and 3.5 (Page 17, Line 705-712) have been modified to include NSCLC trial data.

Comment 17: Discuss hypoxia measurement methods (such as PET imaging, HIF-1α staining) to strengthen translational relevance.

Response: This is outside the scope of this review. 

Comment 18: Limited mechanistic depth on how VEGF signaling directly modulates PD-1/PD-L1 expression.

Response: Section 3.2 has been modified to include this. Page 14, Lines 539-551.

Comment 19: Does not discuss resistance to anti-angiogenic therapies or their limitations (such as toxicity and transient effects).

Comment 20: Limited discussion of overcoming resistance (such as anti-angiogenic failures and microbiome modulation challenges).

Response: The concerns on anti-angiogenic therapies are briefly addressed in sections 3.2 (page 14, Lines 567-572).

Microbiome modulation studies (trials) are still in their infancy. We believe it is premature to discuss strategies to overcome resistance and challenges of microbiome modulation at presence, as these will likely be speculative and extensive.

Comment 21: Lacks recent studies for all sections.

Response: Recent studies have been added to all the sections. Ref 135-137, 140-142, 147, 150-152, 164-166, 173, 179, 180, 188, 190, 192-194, 197, 215-218.

Conclusions:

Comment 22: Highlight clinical implications (such as personalized ICI use based on mutation status). Propose specific future directions (such as combining STK11/KEAP1 testing with TMB, microbiome modulation).

Response: Conclusion has been modified to include these suggestions. Page 18, Line 725-730.

Round 2

Reviewer 3 Report

Comments and Suggestions for Authors

The authors have responded to my comments. I agree for its acceptance.